# Identification of Somatic Structural Variants in Solid Tumors by Optical Genome Mapping

**DOI:** 10.3390/jpm11020142

**Published:** 2021-02-18

**Authors:** David Y. Goldrich, Brandon LaBarge, Scott Chartrand, Lijun Zhang, Henry B. Sadowski, Yang Zhang, Khoa Pham, Hannah Way, Chi-Yu Jill Lai, Andy Wing Chun Pang, Benjamin Clifford, Alex R. Hastie, Mark Oldakowski, David Goldenberg, James R. Broach

**Affiliations:** 1Department of Otolaryngology—Head and Neck Surgery, Pennsylvania State University College of Medicine, Hershey, PA 17033, USA; dgoldrich@pennstatehealth.psu.edu (D.Y.G.); blabarge@pennstatehealth.psu.edu (B.L.); dgoldenberg@pennstatehealth.psu.edu (D.G.); 2Department of Biochemistry and Molecular Biology, Pennsylvania State University College of Medicine, Hershey, PA 17033, USA; schartrand@pennstatehealth.psu.edu (S.C.); lzhang6@pennstatehealth.psu.edu (L.Z.); 3Bionano Genomics, San Diego, CA 92121, USA; hsadowski@bionanogenomics.com (H.B.S.); yzhang@bionanogenomics.com (Y.Z.); kpham@bionanogenomics.com (K.P.); hway@bionanogenomics.com (H.W.); jlai@bionanogenomics.com (C.-Y.J.L.); apang@bionanogenomics.com (A.W.C.P.); bclifford@bionanogenomics.com (B.C.); ahastie@bionanogenomics.com (A.R.H.); moldakowski@bionanogenomics.com (M.O.)

**Keywords:** optical genome mapping, solid tumors, cancer genomics

## Abstract

Genomic structural variants comprise a significant fraction of somatic mutations driving cancer onset and progression. However, such variants are not readily revealed by standard next-generation sequencing. Optical genome mapping (OGM) surpasses short-read sequencing in detecting large (>500 bp) and complex structural variants (SVs) but requires isolation of ultra-high-molecular-weight DNA from the tissue of interest. We have successfully applied a protocol involving a paramagnetic nanobind disc to a wide range of solid tumors. Using as little as 6.5 mg of input tumor tissue, we show successful extraction of high-molecular-weight genomic DNA that provides a high genomic map rate and effective coverage by optical mapping. We demonstrate the system’s utility in identifying somatic SVs affecting functional and cancer-related genes for each sample. Duplicate/triplicate analysis of select samples shows intra-sample reliability but also intra-sample heterogeneity. We also demonstrate that simply filtering SVs based on a GRCh38 human control database provides high positive and negative predictive values for true somatic variants. Our results indicate that the solid tissue DNA extraction protocol, OGM and SV analysis can be applied to a wide variety of solid tumors to capture SVs across the entire genome with functional importance in cancer prognosis and treatment.

## 1. Introduction

One of the hallmarks of cancer is genomic instability, which often affects genes controlling cell division and genome integrity. The resulting alterations include single-nucleotide variant (SNV) point mutations as well as structural variants (SVs), in which larger DNA segments undergo chromosomal perturbations such as deletions, insertions, duplications, inversions, and translocations. For instance, recurrent translocations, such as the Philadelphia chromosome, can activate oncogenes but at the same time reveal avenues for implementing or developing effective targeted drug therapies [1,2,3,4]. Likewise, SV identification plays an increasingly important role in cancer diagnosis and prognosis [5,6], and SVs have been shown to play a crucial role in intra-tumoral genetic heterogeneity [7]. Therefore, SV identification and analysis are important to understanding oncogenesis and tumor behavior.

Short-read sequencing can readily detect many SNVs, but is less successful in detecting SVs, by either alignment-based or assembly-based methods [8]. Since alignment-based approaches rely on mapping reads to unique positions, repetitive and low-complexity genomic regions can lead to misalignment and false-positive SV calls. Additionally, homologous alleles may be incorrectly combined, leading to haploid assembly only representing a single allele or chimeric assemblies mixing alleles. Whole-genome and cytogenetic approaches such as whole-genome sequencing (WGS), karyotyping, fluorescent in situ hybridization (FISH) and CNV microarrays also contain significant limitations. Karyotyping provides a comprehensive view of the entire genome but carries limited resolution of ~5 Mb and in most cases requires culturing cells before preparing chromosomes. FISH has a higher resolution but requires prior knowledge as to which loci to test and has limited throughput. CNV microarrays offer a resolution down to multiple Kb but are insensitive to balanced chromosomal aberrations such as translocations and inversions, are unable to detect low-frequency allelic changes, and cannot distinguish tandem duplications from insertions in trans. Finally, WGS has difficulty with de novo genome assembly and resolving duplications and repeated sequences [8,9,10]. Therefore, alternative methods are required to preserve long-range genomic structural information.

Optical genome mapping (OGM) has emerged as a viable option for analyzing large genomes for SVs. OGM preserves long-range information by imaging entire intact molecules of DNA in their native state and, as a result, has contributed to constructing reference genome assemblies, including those for maize, mouse, goat, and humans [11,12,13,14,15,16,17,18,19,20,21,22,23,24,25,26,27,28]. OGM can detect large (>500 bp) and complex SVs, such as chromothrypsis, that are difficult to detect using traditional short-read sequencing alone. OGM preparation and analysis workflow has been successfully applied to liquid-phase tumor and cell culture SV analyses. For instance, investigators have analyzed primary leukemic cells with OGM to identify previously unrecognized SVs implicated in oncogenesis and patients’ survival and have combined OGM with chromosome conformation capture to demonstrate enhancer highjacking resulting from SVs [5,29,30]. Similarly, investigators used OGM to visualize complex gene fusions and novel somatic SVs in liposarcoma, melanoma and other well-studied cancer cell lines [31,32].

Despite its success in visualizing SVs in liquid tumors and cell lines, OGM has not yet seen widespread application in solid tissue tumors, due primarily to the difficulty of obtaining high-quality, high-molecular-weight DNA from solid tumor samples. Nonetheless, previous work has shown the feasibility of high-quality high-molecular-weight DNA isolation and analysis using earlier workflow iterations [33], and recent feasibility studies have shown the importance of OGM application to solid tumor analysis [7,34,35]. Peng et al. demonstrated large SVs not detected by WGS implicated in metastatic lung squamous cell carcinoma [7], and Jaratlerdiri et al. and Crumbaker et al. similarly found SVs impacting oncogenic and tumor-suppressing genes not identified by NGS or WGS alone in prostate cancer [34,35]. However, these previous methods for extracting gDNA from solid tissue were either prohibitively expensive or yielded low quantities of DNA [36]. We demonstrate here the successful implementation of a workflow to generate ultra-high-molecular-weight gDNA and subsequent SV analysis for 20 solid tumor samples comprising a wide variety of solid tissue organ systems.

## 2. Materials and Methods

### 2.1. Tumor Samples

Solid tissue was collected following surgical resection for 10 tumors: four squamous cell carcinomas of the tongue, three anaplastic carcinomas of the thyroid, one liver hepatocellular carcinoma, one lung pleomorphic carcinoma, and one bladder tumor. Patients consented under protocols approved by the Penn State Health Institution Review Board and tissue was flash frozen and stored at −80 °C in the Penn State Institute for Personalized Medicine (IPM). Ten additional fresh frozen solid tumor samples were acquired from BioIVT for the following tumor types: lung adenosquamous carcinoma, liver hepatocellular carcinoma, bladder papillary urothelial carcinoma, kidney renal cell carcinoma, breast ductal carcinoma in situ, prostate invasive adenocarcinoma, brain anaplastic astrocytoma, ovarian serous carcinoma, colon adenocarcinoma, and papillary thyroid carcinoma. For some of the samples, two or three separate sections of the tumor were excised and processed independently to provide duplicate or triplicate biological replicates.

### 2.2. Bionano Optical Genome Mapping

*Ultra-High-Molecular-Weight gDNA Isolation from Solid Tissue.* The following protocol is diagrammed in Figure 1 and described in greater detail in a support document from Bionano Genomics (https://bionanogenomics.com/support-page/sp-tissue-and-tumor-dna-isolation-kit/). Briefly, tissue sections with a target mass of 10 mg were sliced from a frozen parent piece on a sterilized aluminum block over dry ice. The tissues were minced briefly and placed into a 15 mL conical tube on ice containing homogenization buffer (HB) for subsequent blending with a Tissueruptor II (Qiagen). Following tissue disruption, samples were washed in additional HB, poured through a 40 μm filter, and centrifuged to pellets, from which the supernatants were decanted. 

Pellets were resuspended in Wash Buffer A (Bionano, San Diego, CA, USA) and transferred to microcentrifuge tubes for additional washing. Supernatants were then decanted, and pellets resuspended in residual volume. Proteinase K (Bionano Genomics, San Diego, CA, USA) was added to samples, followed by Lysis and Binding Buffer (LBB, Bionano Genomics, San Diego, CA, USA) and mixed to produce a lysate containing high-molecular-weight DNA. Phenylmethylsulfonyl Fluoride Solution (PMSF, Millipore Sigma) was added to inactivate Proteinase K, followed by Salting Buffer (SB, Bionano Genomics, San Diego, CA, USA).

A single paramagnetic Nanobind Disc (Bionano Genomics, San Diego, CA, USA) was added to the lysate with 100% isopropanol, to facilitate binding and washing of gDNA strands. With gDNA captured on the disc, the supernatants were carefully removed and discs were washed with rounds of ethanol-based wash buffer. Discs were then transferred to clean tubes, where gDNA was eluted in buffer and homogenized at room temperature. 

*Ultra-High-Molecular-Weight gDNA Isolation from Blood.* Previously frozen EDTA-stabilized blood aliquots were thawed, inverted to mix, and measured for white blood cell counts (HemoCue, Brea, CA USA, WBC). Blood volumes corresponding to 1.5 × 10^6^ cells were transferred to a microcentrifuge tubes, then spun to obtain cell pellets. After removing supernatants, pellets were resuspended in 40 μL Stabilizing Buffer and 50 μL Proteinase K (Bionano Genomics, San Diego, CA, USA). Lysis and Binding Buffer (LBB, Bionano Genomics, San Diego, CA, USA) was then added and mixed to produce a lysate, after which isolation of DNA was performed essentially as described above for tumor tissue.

*Direct Label and Staining (DLS).* For both tumor- and blood-derived samples, gDNA was labeled in Direct Label and Stain reactions, in which fluorescent labels are enzymatically conjugated to a six-base pair recognition sequence followed by DNA counterstaining. Briefly, 750 ng gDNA was diluted and mixed with a labeling master mix containing DLE-1 Enzyme and DL-Green (Bionano Genomics, San Diego, CA, USA). Reactions were shielded from light and incubated at 37 °C for 2 h. A Proteinase K solution then inactivated the enzyme, and successive membrane adsorption steps were used for cleanup. A portion of each sample was then carried forward into a staining master mix addition, slowly homogenized, and incubated overnight at room temperature.

The DNA concentration of each labeled sample was confirmed within 4–12 ng/µL by High-Sensitivity dsDNA Qubit Assay and then loaded onto a Bionano Saphyr^®^ Chip (Bionano Genomics, San Diego, CA, USA, Part#20366) and run on the Bionano Saphyr^®^ instrument, targeting approximately 300× human genome coverage. 

### 2.3. Bionano Access and Solve Pipeline

Genome analysis was performed using Rare Variant Analysis in Bionano Access 1.6 and Bionano Solve 3.6, which captures somatic SVs occurring at low allelic fractions. Briefly, molecules of a given sample dataset were first aligned against the public Genome Reference Consortium GRCh38 human assembly. SVs were identified based on discrepant alignment between sample molecules and GRCh38, with no assumptions about ploidy. Consensus genome maps (*.cmaps) were then assembled from clustered sets of at least three molecules that identify the same variant. Finally, the genome maps were realigned to GRCh38, with SV data confirmed by consensus forming final SV calls. SVs were then annotated with known canonical gene set present in GRCh38, as well as estimated population frequency for each structural variant detected by comparing to a custom control database (*n* = 297) from Bionano Genomics.

### 2.4. Data Comparison

Whole-genome imaging data were compared to the human reference genome GRCh38 (hg38) to retain only those SVs not present in the reference genome. SVs were further filtered to eliminate any variant observed in any of the Bionano control samples or, if available, patient-matched blood. Bionano Access-created csv files containing filtered SVs were analyzed to compare SV content across samples. For tissue samples with associated blood samples, control database filtration efficacy was compared to blood-filtering efficacy at identification of somatic mutations. For duplicate/triplicate samples, filtered SVs were compared to determine intra-sample reliability. For identification of cancer-related genes, the set of genes affected by SVs in each of the samples was compared to the list of genes causally implicated in cancer available in the Cosmic Cancer Gene Census database (v92) [37] (https://cancer.sanger.ac.uk/census). 

## 3. Results

*Patient Clinical Characteristics.* Clinical data for the patients from whom tumor samples were acquired are shown in Table 1. A total of 60% (12/20) patients were male, with a mean age of 73.5 years at sample acquisition. A total of 45% (9/20) patients identified as Caucasian, 40% (8/20) as Asian, and 5% (1/20) as Hispanic, with 10% (2/20) not identifying. The majority of IPM-sourced tumor samples were obtained from Caucasian patients (7/10), while the majority of the BioIVT-sourced tumor samples were obtained from patients of Asian ethnicity (8/10). In terms of overall risk factors, 55% (11/20) of patients were self-described current or former tobacco users and 45% (9/20) endorsed some history of alcohol use. 

The tumor samples consisted of a variety of stages (Table 1). A total of 75% (3/4) of tongue cancer samples and 100% (3/3) anaplastic thyroid cancers were stage IV cancers, while 100% (2/2) lung and (2/2) bladder cancers were stage II. Limited tumor data were available for the commercially available BioIVT-sourced tumor samples. 

*DNA Quality Metrics*: All 20 solid tumors yielded high-molecular-weight gDNA (Table 2). The average concentration across all samples following gDNA isolation was 120 ng/µL by Broad Range dsDNA Qubit Assay. All eluted gDNA were well above the minimal concentration required for DLS labeling (35 ng/µL) and the average final DNA yields for each tumor ranged from 1.2 to 16.4 µg/10 mg input tissue. Analysis on a Saphyr instrument following DLS labeling revealed that samples achieved an average label density of 14.4/100 Kbp, average filtered N50 (>20 Kbp) DNA size of 242 Kbp, average filtered N50 (>150 Kbp) DNA size of 315 Kbp, map rate of 82.62%, effective reference coverage of 320× and average effective DNA throughput (≥150 Kbp) of 50 Gbp/scan. Rare Variant Pipeline Analysis of the samples yielded an average of 82.4% of molecules aligning to the reference genome. These values are all well above the acceptable range for obtaining high-quality data and none of the samples failed any of these quality control metrics.

*Identification of somatic structural variants*. Rare Variant analysis of the samples revealed a large numbers of variants in each sample, only a fraction of which were likely somatic. The unfiltered analysis yielded an average of 1633 total SVs per sample (range 1241–2000), which include both somatic and germline polymorphic variants (Figure 2, upper panel). These consisted predominantly of insertions and deletions, with an average of 712 insertions and 604 deletions, a fewer number of inversion (an average of 153) and duplications (an average of 123), and relatively few translocations (an average of 41). Eliminating those SVs found in Bionano’s control database of known polymorphic SVs reduced the number of putative somatic structural mutations by 91% to an average of 124 total SVs per sample (Figure 2, lower panel). Most of the variants eliminated were insertions and deletions, of which on average 97% and 94%, respectively, were removed. On the other hand, less than 0.2% of the translocations were flagged as polymorphic, consistent with the fact that almost no translocations persist in the population as polymorphisms.

To determine the efficacy of identifying somatic SVs by filtering against Bionano’s database of known polymorphisms, we used as a gold standard the blood samples from four patients from whom we had obtained tongue tumors. That is, we determined the true somatic mutations in each of these four tumors by eliminating those SVs identified in each of the tumors that were also present in the corresponding blood sample. We could then compare those true somatic variants to the list of somatic variants predicted by filtering against the database of polymorphisms. For these four tongue tumor samples, we identified an average of 1474 total SVs per sample. Filtering these SVs using the Rare Variant Analysis pipeline for SVs not found in the Bionano control database yielded an average of 72 total SVs per sample, consisting of 11 insertions (range 9–15), 31 deletions (range 11–47), 3 inversions (range 1–6), 14 duplications (range 2–23), and 14 translocations (6–19) (Figure 3, right upper panel). Filtering against the variants found in the corresponding blood samples returned an average of 58 total SVs per sample, consisting of 10 insertions (range 9–10), 20 deletions (range 7–35), 2 inversions (range 0–4), 13 duplications (range 4–24), and 14 translocations (range 6–19) (Figure 3, left upper panel). 

Comparing the residual SV sets obtained by filtering against Bionano’s control database to the sets of true somatic SVs for each sample demonstrated that the control database filtration exhibited strong statistical accuracy (Figure 3, lower panel). Across the four separate samples, the control database exhibited an average sensitivity of 92% (83–96%) and specificity of 98% (range 97–99%). That is, filtering with the control database retained most of the true somatic mutations while eliminating almost all of the polymorphic SVs. Similarly, the average negative predictive value of the filter was 99.6%, demonstrating that an SV identified as germline was indeed a germline variant, while the positive predictive value of 74% (range 60–81%) indicates that a majority, but not all, the variants identified as somatic are in fact somatic. In other words, the results obtained by filtering SVs against Bionano’s control database retained almost all the true somatic mutations. However, several of the SVs identified as somatic were actually germline. Those SVs inaccurately identified as somatic were rare germline variants, predominantly insertions or deletions, essentially private to the patient’s genome. As above, we noted that the filtering process did not affect all SV types equally: while most deletions and insertions were flagged as polymorphic and eliminated from the list of somatic mutations, very few duplications and essentially no translocations were identified as polymorphic. This is consistent with observation that few translocations or duplications are stable through meiosis. 

*Duplicate Sample Analysis*. We compared SV calls from separate isolates of the same sample to assess consistency and reproducibility of the method, albeit without knowing the extent of tumor heterogeneity of the individual samples. Six samples underwent triplicate analysis, and four samples underwent duplicate analysis (Table 3). After identifying SVs using the Rare Variant Analysis pipeline and filtering them against the Bionano control database of known polymorphisms, we recovered an average of 116 somatic SVs shared among the separate isolates of the same tumor. These comprised an average of 23 insertions, 29 deletions, 10 inversions, 11 duplications and 43 translocations (Table 3). As noted above, the number of SVs identified in a tumor varied widely across the different tumors examined, with lung, breast, brain and ovarian tumors showing a high level of somatic SVs while the others containing a relative low number of SVs. Moreover, the percentage of SVs shared among different isolates of the same tumor also varied among the different tumor types. However, the percentage of shared SVs and the total number of SVs were uncorrelated. Assuming that the higher values for shared SVs reflect the reproducibility of the method, then we might postulate that the lower shared values represent both the reproducibility and the tumor heterogeneity. That is, we would suggest that the reproducibility of the method across multiple biological replicates is 85–95%, corresponding to the values obtained from those samples with the least variability. Thus, we would suggest that the residual variability in those samples with lower reproducibility (50–75%) reflects heterogeneity of SVs in the tumors. This would suggest that these brain, liver, lung and prostate tumors had a relatively high level of tumor heterogeneity. 

The number and types of somatic variants in a tumor varied substantially across the collection of samples (Figure 4). Several tumor samples, including those from colon, bladder, kidney and all four from thyroid, contained relatively few somatic SVs whereas others, including those from prostate, ovaries, lung and brain, carried a large number of somatic SVs. Since these samples for the most part serve as single representatives of each tumor type, we cannot extrapolate to the tumor types as a whole the contribution of SVs to cancer onset and development for each class of tumor. However, it is noteworthy that the SNV mutational burden in thyroid cancers is among the lowest among all tumor types and that measure of genome instability is mirrored in the low number of somatic SVs in all four of the samples examined [39]. Similarly, the SNV mutational burden in lung cancers is among the highest across all tumor types and both of the lung tumors examined here also carry a high level of somatic SV. Finally, the extent of somatic SVs observed in our collection of tumors does not correlate with either cancer stage nor with obvious lifestyle characteristics (Table 1). For instance, neither smoking nor drinking history has a stronger influence on SV mutation burden than does site of origin of the tumor. However, further data examining the correlation of lifestyle characteristics and tumor stages with SV mutational burden are warranted to assess the impact of these behaviors on SV formation and persistence.

*Identification of Cancer Gene Mutations*. While, as noted above, we cannot generalize regarding the role of structural variants in onset and progression of different tumor types, our results indicate that we can extract from the structural variant list clinically relevant data on individual tumors that might inform prognosis or treatment options. We examined the somatic structural variants in each tumor sample for those that affected genes previously associated with cancer. In particular, we annotated those genes altered by a structural variant, either by disruption, duplication, deletion or fusion, and intersected that list with the set of cancer-related genes in the Cosmic database (v92) [37]. The resultant list by tumor type is provided in Table 4 and subdivided into oncogenes, tumor suppressor genes and gene fusions. We included only those oncogenes that were potentially activated by duplication or gene fusion and only those tumor suppressor genes that were potentially inactivated by deletion, insertion or fusion. As evident, every tumor sample carried at least one such cancer gene mutation and most contained multiple hits. Several of these genes offer the opportunity for targeted therapies, focused either directly on the oncogene, as would be the case for CDK6 and ERBB2, or at the pathway downstream of the affected gene, as would be the case for BRAF and CDKN2A. Other affected genes, such as MSH2, RAD51B, RAD21 and RAD18, suggest the potential of therapy based on possible ensuing genome instability, such as immunotherapy or PARP inhibitors. Many of these variants would not be readily identified by targeted gene panels generally used for clinical assessment of tumor genomes. Moreover, in many cases, the cancer genes altered by SVs were not previously associated with the cancer type in which we observed it. For instance, we observed a fusion of CDK6 in one of the tongue tumors while it has previously been associated predominantly only with ALL. Similarly, LRP1B is often inactivated in CLL or ovarian cancer, while we find it inactivated by deletion in one of the lung tumors. Thus, the identification of somatic structural variants by OGM could provide useful clinical insights not readily available through standard next-generation sequencing or targeted panels.

Diagrams of somatic structural variants in all the solid tumor genomes, filtered to remove known polymorphisms, showing translocations and inversions in the center, copy number on the inner ring and insertions (green), deletions (orange) inversions (light blue) and duplications (violet) on the next to most outer ring. Chromosomes are ordered sequentially in a clockwise orientation in the outer ring on which are indicated cytological banding patterns and the centromere (red bar).

In addition to identifying individual cancer-related genes in tumor types, our results provide a panoramic view of the entire tumor genome and reveal large-scale genomic features not readily available from standard sequencing techniques. As evident in the results in Figure 4, our data provide a rapid snapshot of the extent of genomic instability in each of the tumors. Such images present an integrated picture of the aneuploidies, translocations, inversions, deletions and insertions, which offers a readily digestible impression of the extent of genetic instability underlying a tumor. Moreover, several large-scale features are evident in these data. For instance, chromothripsis is a massive cluster of chromosomal rearrangements localized to a restricted region of a chromosome, which often results from a single catastrophic event [40]. Figure 5 details a chromothripsis event on a portion of chromosome 5 in one of the lung tumor samples. In fact, such events are readily evident in four of the Circos plots in Figure 4, consistent with previous estimates of 2–3% prevalence across all cancers, albeit with different frequency in different cancers [41]. The detection and mapping of such a feature are difficult to achieve by short-read sequencing [41] but can indicate poor prognosis and the corresponding need for aggressive therapy. 

## 4. Discussion 

In this report, we described the application of optical genome mapping to solid tumors, which we suggest can significantly augment the genomic analysis of such tumors obtained by next-generation sequencing. Genomic analysis of tumors has stimulated major advances in cancer diagnosis, prognosis and treatment, shifting the focus from morphological and histochemical characterization to consideration of the landscape of driver mutations in the tumor [42,43,44]. Somatic driver events in a tumor—point mutations and structural variants (SVs) including insertions, deletions, inversions, translocations and copy number changes—are currently identified in solid tumors by some combinations of RNA sequencing and genome sequencing of either targeted gene panels, whole exomes or whole genomes. As noted in this report, OGM can provide a pervasive view of the structural variants in a tumor and the cancer-related genes on which they impinge, thus identifying affected genes agnostically, without prior bias imposed by gene panels.

Some prior studies have begun to demonstrate the utility of Bionano DNA isolation protocols in solid tissue tumor analysis. These include studies of lung squamous cell carcinoma and metastatic prostate carcinoma [7,34,35]. This current report demonstrates the utility of the DNA isolation protocol and SV analysis in a wide variety of solid tissue types, and expands the feasibility of such analysis for previously unused human tissue types. The high DNA yield, high effective coverage, map rate and other molecular quality metrics shown across tumor types confirm how our extraction and analysis workflow can be effectively applied to many solid tissue tumors. 

This current DNA isolation protocol carries a number of advantages. Tissue handling can be performed at room temperature. The current protocol showed successful DNA isolation in solid tissue samples of <20 mg, and even as low as 6 mg. The low tissue input requirement carries important applications for rare cancer samples, human tissue biopsy testing and other low-quantity specimen acquisition. Additionally, utilizing the novel paramagnetic Nanobind disks rather than prior agarose gel plugs greatly decreases time needed to complete DNA isolation to only 5 h. The ability to isolate DNA from up to eight simultaneous samples using the current protocol greatly amplifies throughput and reduces tissue-to-data processing time, increasing both laboratory convenience as well as expanding potential for clinical utility where rapid data turnaround is paramount. Furthermore, the strong inter-sample SV correspondence shown by most tissue types in duplicate/triplicate sample analysis demonstrates the reproducibility of this technique; intra-sample heterogeneity of select samples may be attributed to non-tumor normal tissue within some tissue fragments, or attributed to specific cancer subtype, and merits further investigation. Although the isolation protocol described here affords many advantages, there are some limitations to this protocol. While high-quality DNA isolation and OGM SV analysis was obtained for a wide variety of tumor types that were tested, it may not be generalizable to every additional untested solid tumor type. Future directions include continuing to validate this protocol in additional tissue types, and assessing additional tumor samples to assess broader trends in the role of specific OGM-identified SVs in individual cancer subtypes.

In clinical evaluation of liquid tumors such as leukemia, genomic analysis is augmented by karyotyping, which gives a panoramic, albeit low resolution, view of the entire genome. Despite the low resolution, the genome wide view of the structural changes afforded by karyotyping reveals diagnostic features of the tumor that have strong prognostic value. Given the consistent correlation of clinical outcomes with specific mutation classes, the World Health Organization (WHO), National Comprehensive Cancer Network (NCCN) and European Leukemia Net (ELN) agencies developed recommendations for diagnosis and management of acute myeloid leukemia in adults based on the spectrum of somatic point mutations and SVs generally revealed by karyotyping [45]. SVs, particularly translocations and inversions, are major considerations in this diagnosis. Since karyotyping is a very challenging technique to apply to solid tumors, the clinician does not have access to a comparable global view of a solid tumor’s genome and the role of SVs in prognosis has likely been underappreciated. Applying OGM broadly to cancer types and correlating SVs revealed by that analysis with clinical outcomes could provide new genomic markers for prognosis and treatment selection.

## 5. Conclusions

We demonstrate the utility of a DNA isolation protocol for high-molecular-weight DNA extraction and OGM SV analysis of a wide variety of solid human tumor types on the Bionano Saphyr system, including breast, colon, liver, brain, bladder, kidney, lung, ovary, prostate and thyroid cancer tissue. The system can be used to accurately detect genetic mutation hallmarks in cancer tissue samples, including rearrangements such as translocations, gene fusions and copy number alterations. Somatic SVs can be determined by comparison filtering with the Bionano control sample database, or against a matched pair sample. Importantly, Bionano SV pipelines can detect SVs with complex breakpoint structures that are difficult to detect with other technologies. Our results indicate that the solid tissue DNA extraction protocol can be applied to a wide variety of solid tumors, and that the Saphyr system can capture, in a streamlined workflow, a broad spectrum of SVs. These SVs have functional importance and provide great utility in cancer prognosis and treatment.

## Figures and Tables

**Figure 1 jpm-11-00142-f001:**
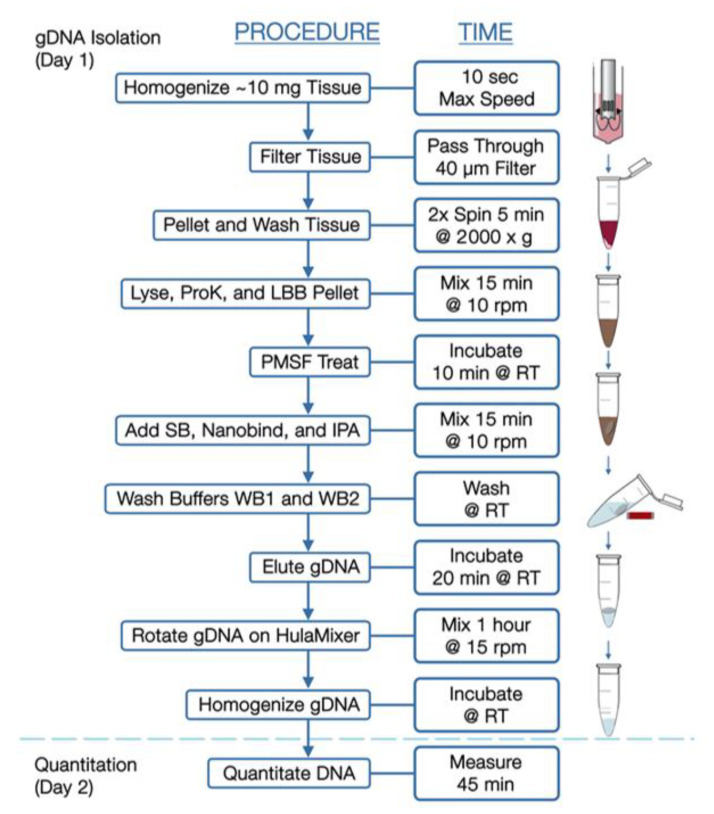
Workflow for isolation of high-molecular-weight DNA from solid tumors.

**Figure 2 jpm-11-00142-f002:**
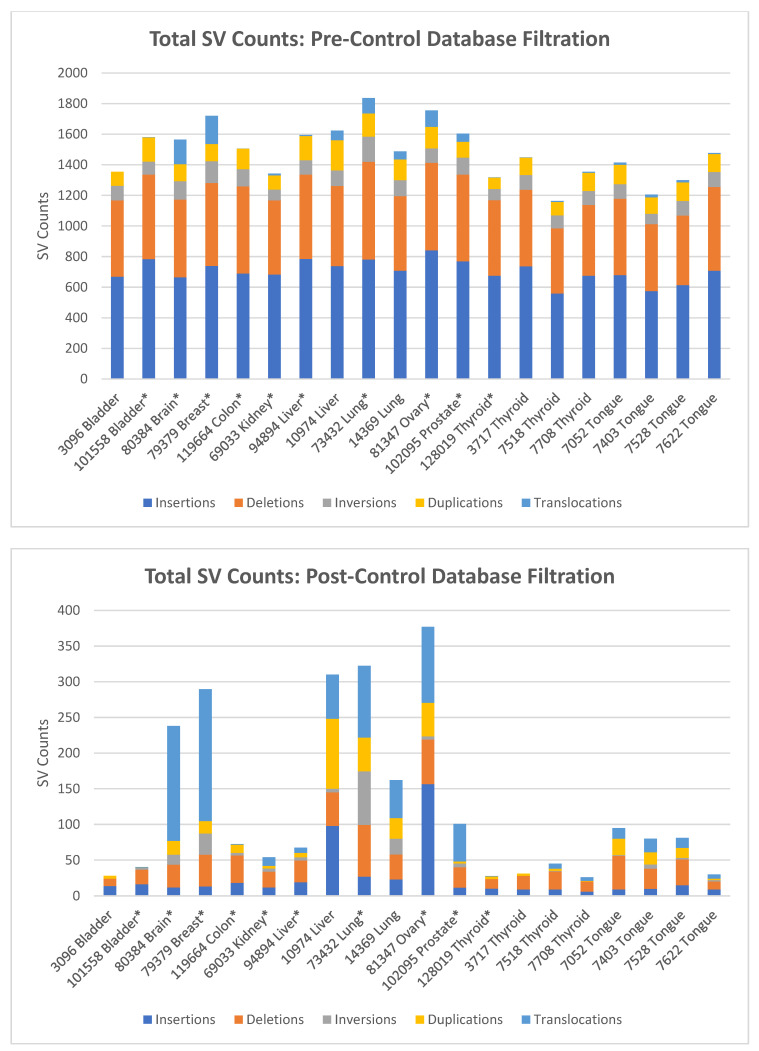
Total and somatic structural variants present in tumor samples. Upper panel: SV counts as determined using the Bionano Rare Variant pipeline, before control database filtration. SV counts are averages for duplicate and triplicate samples. Lower panel: SV counts after filtering total SVs to remove known polymorphic SV found in Bionano’s GRCh38 control database. SV counts are averages for duplicate and triplicate samples, which are indicated by (*).

**Figure 3 jpm-11-00142-f003:**
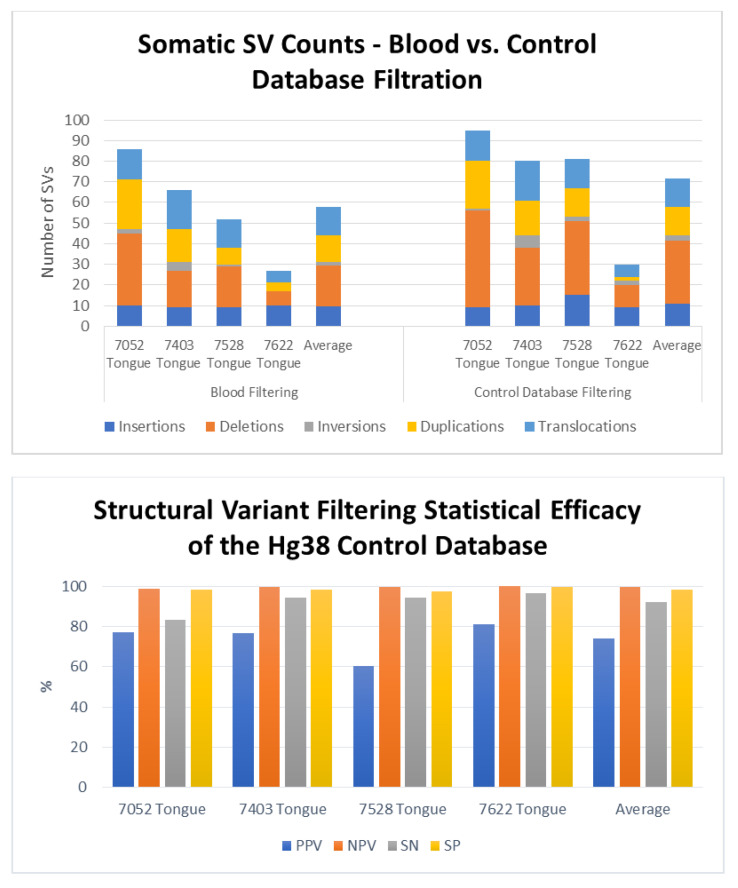
Efficacy of the somatic variant identification using a control database of known polymorphisms. Upper Panel: Number and distribution of somatic structural variant in four tongue tumors as determined by filtering against SVs in the patient’s genome from peripheral blood (left) or against Bionano’s control database of known polymorphisms. Lower Panel: Values for sensitivity (SN), specificity (SP) and positive (PPV) and negative predictive values (NPV) for identification of somatic structural variants obtained by filtering total identified SVs to remove those present in a control database of know human polymorphisms. Data obtained by filtering against the control database were compared to those obtained by filtering total SVs to remove those present in the genomes obtained from peripheral blood from the each of the patients from whom the tumors were removed.

**Figure 4 jpm-11-00142-f004:**
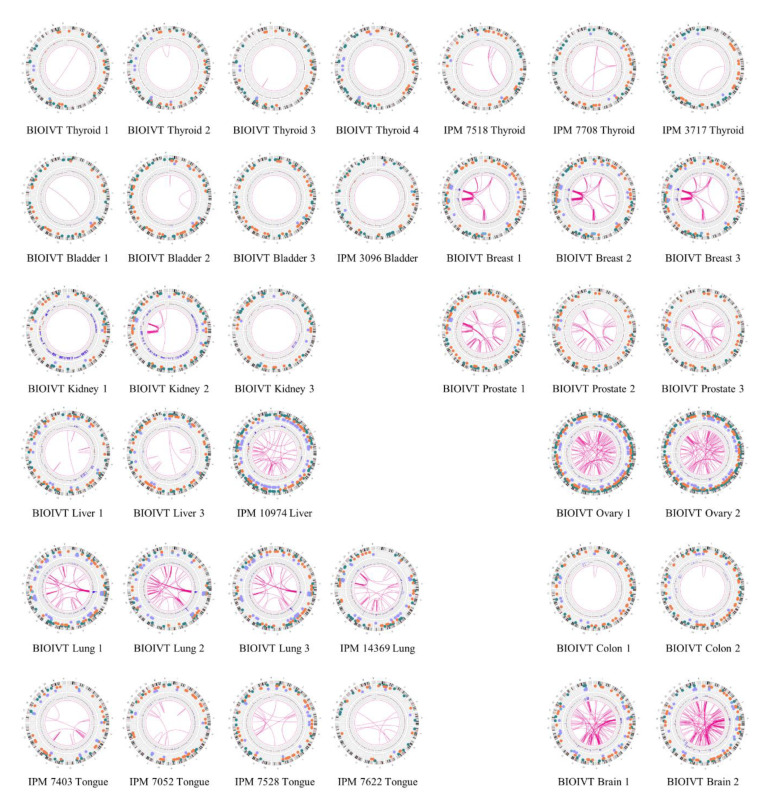
Global view of structural variants in solid tumor samples. Diagrams of somatic structural variants in all the solid tumor genomes, filtered to remove known polymorphisms, showing translocations and inversions in the center, copy number on the inner ring and insertions (green), deletions (orange) inversions (light blue) and duplications (violet) on the next to most outer ring. Chromosomes are ordered sequentially in a clockwise orientation in the outer ring on which are indicated cytological banding patterns and the centromere (red bar).

**Figure 5 jpm-11-00142-f005:**
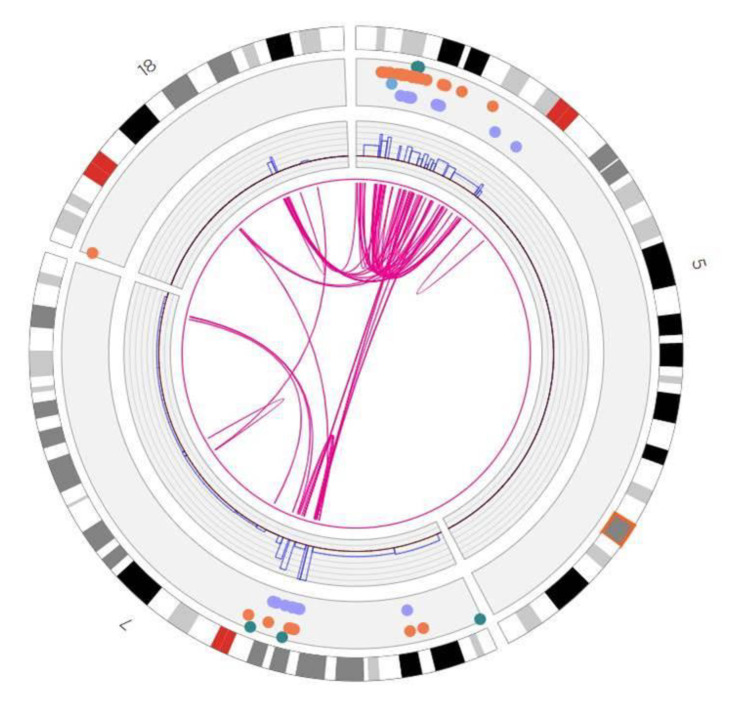
Chromothrypsis of chromosome 5p in a lung tumor. Shown is a truncated Circos plot of the lung tumor, focused on the region of chromosome 5, highlighting the chromothrypsis event that occurred on its p arm. The organization of the Circos plot is as indicated in the legend to Figure 4.

**Table 1 jpm-11-00142-t001:** Patient demographics and tumor characteristics.

Study ID	Cancer Type *	Age ^†^	M/F	Ethnicity	Smoking History	Alcohol History	Pathologic TNM ^‡^	Cancer Stage
7528	Tongue (SCC)	25	M	Caucasian	None	Rare	T3N2bM0	IVa
7052	Tongue (SCC)	35	M	Caucasian	None	None	T2N3M0	IVb
7622	Tongue (SCC)	60	F	Caucasian	50 pack years	1–2 drinks/week	T3N0M0	III
7403	Tongue (SCC)	65	M	Caucasian	45 pack years	Rare	T2N3bM0	IVb
7518	Thyroid (AP)	70	F	Caucasian	20 pack years	2 drinks per day	T4bN1bM1	IVc
7708	Thyroid (AP)	65	M	Caucasian	None	None	4aN1bM1	IVc
3717	Thyroid (AP)	80	M	Hispanic	25 pack years	Rare	T4aN1aM1	IV
14369	Lung (pleomorphic carcinoma)	60	M	N/A	60 pack years	None	T2bN1M0	IIa
10974	Liver (metastic adenocarcino-ma of colon)	65	F	N/A	Former	None	T3N2aM1	IVB
3096	Bladder (urothelial carcinoma)	55	M	Caucasian	60 pack years	None	T2N0M0	II
73432	Lung (adeno-squamous carcinoma)	35	M	Asian	Former (5 pack years)	Former (1 per day, 10 years)	T2aN1M0	IIA
94894	Liver (hepato-cellular carcinoma)	70	M	Asian	7 pack years	1 per day, 35 years	T1NxM0	I
101558	Bladder (papillary urothelial carcinoma)	65	M	Asian	Former (5 pack years)	1 per day, 20 years	T2NxM0	II
69033	Kidney (renal cell carcinoma)	60	F	Asian	None	None	T2bNxM0	II
79379	Breast (ductal carcinoma in situ)	50	F	Asian	None	None	T3N0M0	IIB
102095	Prostate (invasive adeno-carcinoma)	60	M	Caucasian	40 pack years	None	T3bN1M0	IV
80384	Brain (anaplastic astrocytoma)	40	F	Caucasian	None	None	NA	NA
81347	Ovarian (serous carcinoma)	75	F	Asian	None	None	T1aN0M0	IA
119664	Colon Cancer (adenocarcinoma)	80	M	Asian	2 pack years	1 per day, 40 years	TXNXMX	UNK
128019	Thyroid (papillary)	35	F	Asian	None	None	T3bNxM0	I

* SCC: squamous cell carcinoma; AP: anaplastic.^†^ ~Age (≥Age-3 and ≤Age+3) ^‡^. Pathologic Staging: Tumor, Node Metastasis (TNM) staging is the internationally accepted system set forth by the American Joint Committee on Cancer (AJCC) used to determine cancer disease stage and guide prognosis and treatment (https://www.cancerstaging.org) [38].

**Table 2 jpm-11-00142-t002:** Single-molecule quality report metrics.

Tissue	No. of Duplicates	Input (mg)	DNA (ng/µL)	DNA Yield (µg/mg)	N50 Kbp (>20 Kbp)	N50 Kbp (>150 Kbp)	Labels/100 kbp	Map Rate (%)	Gbp/Scan	Effective Coverage
7528 (tongue)	1	17.5	37	0.12	211	317	12.3	58.8	53	237×
7052 (tongue)	1	17.1	81	0.28	179	287	15.2	82.4	37	345×
7622 (tongue)	1	18.7	160	0.51	315	361	13.4	75.8	64	317×
7403 (tongue)	1	18	79	0.26	148	272	14.8	72.8	33	304×
7518 (thyroid)	1	8.6	28	0.20	143	265	14.4	76.6	26	312×
7708 (thyroid)	1	10.6	85	0.47	269	356	13.1	61.2	35	253×
3717 (thyroid)	1	13.2	49	0.22	250	320	14.5	88.2	58	371×
14369 (lung)	1	11.4	87	0.45	268	323	14.0	89.6	36	372×
10974 (liver)	1	6.5	82	0.74	235	289	15.2	87.9	49	360×
3096 (bladder)	1	9.4	59	0.37	265	319	13.8	78.3	39	325×
73432 (lung)	3	9.6	128	0.86	248	304	15.0	90.4	51	339×
94894 (liver)	2	9.0	196	1.41	265	306	14.9	89.3	84	325×
101558 (bladder)	3	9.7	245	1.64	313	357	15.2	91.8	66	338×
69033 (kidney)	3	10	96	0.63	201	269	14.6	83.5	41	296×
79379 (breast)	3	13.3	183	1.04	317	395	14.2	84.1	77	288×
102095 (prostate)	3	10.3	113	0.72	273	361	14.8	85.1	62	295×
80384 (brain)	2	10.5	168	1.06	228	292	14.6	90.2	42	306×
81347 (ovary)	2	10.5	168	1.05	228	292	14.6	90.2	42	330×
119664 (colon)	2	11.3	231	1.33	263	330	14.9	88.6	42	274×
128019 (thyroid)	4	10	126	0.77	213	294	14.5	87.6	64	294×
Average	1.9	11.8	120.	0.71	241.	315.	14.4	82.6	50.0	314×

Average values are presented for samples with multiple replicates.

**Table 3 jpm-11-00142-t003:** Duplicate Sample Analysis. Shown are the number of somatic structural variants shared among the multiple isolates of the same sample and the percentage of those relative the total number of somatic variants found in all the isolates of the same sample.

	Total	%	Insertion	%	Deletion	%	Inversion	%	Duplication	%	Translocation	%
Brain *	134	70	5	63	21	78	9	69	13	72	86	69
Colon *	63	93	15	83	36	97	2	100	9	90	1	100
Liver *	45	70	14	74	21	81	1	17	4	67	5	71
Ovary ^‡^	338	86	136	82	59	87	4	80	40	85	99	91
Bladder ^‡^	30	88	11	79	18	100	1	100	0	100	0	0
Breast ^‡^	221	92	9	82	33	85	23	88	14	88	142	95
Kidney ^‡^	19	76	6	67	11	85	2	100	0	0	0	100
Lung ^‡^	221	66	18	69	53	75	59	73	26	50	65	63
Prostate ^‡^	69	48	8	47	22	61	3	38	1	25	35	44
Thyroid ^‡^	19	86	7	88	10	91	0	100	2	100	0	0
Average (all)	116	78	23	73	28	84	10	76	11	68	43	63
Duplicate Average	145	80	43	75	34	86	4	66	17	78	48	83
Triplicate Average	97	76	10	72	25	83	15	83	7	60	40	50

* = duplicate sample, ‡ = triplicate sample.% = % of SV calls shared among duplicate/triplicate samples.

**Table 4 jpm-11-00142-t004:** Structural variants affecting cancer relevant genes.

Sample	Oncogene	Tumor Suppressor	Gene Fusion
Prostate	ERBB2 (Dup)	PTEN (Del)	PTEN-LINC01374
	GATA2 (T)	NF1 (Del)	DHX30-GATA2
	NUP98 (T)		CASC15-NUP98
			PRKAR1A-FRMPD4
			ERG-TMPRSS2
			FREM1-MYH9
Ovarian		NUMA1 (T)	NBEA-ZFHX3
		NF1 (I)	HMGN2P46-BLOC1S6
		SMARCA4 (I)	LPP-PIEZO1
Kidney	PRKAR1A (T)	CDKN2A (Del)	PRKAR1A-FRMPD4
	ERBB2 (Dup)	ZFHX3 (Del)	
Colon		FHIT (Del)	
Breast	ERBB4 (Dup)	USP8 (T)	USP8-PRPSAP
	ERBB2 (Dup)	PRKAR1A (T)	PRKAR1A-FRMPD4
		RAD51B (Del)	LINC01476-BRIP1
		CDKN2A (Del)	SYK-CFAP77
Brain	SETBP1 (T)	LARP4B (T)	CCDC158-LARP4B
		CSMD3 (T)	CA13-CSMD3
		LRP1B (Del)	DPYD-SETBP1
		RAD21 (Del)	CNBD1-AC083836.1
Bladder		DDX10 (Del)	
Tongue	BRAF (T)	CDKN2A (Del)	EPHB1-BRAF
	CDK6 (T)	PTPRD (Del)	CDK6--AC091551.1
	CCND2 (T)	RAD51B (T)	PCLO-RAD51B
	CCND1 (Dup)	LRP1B (Del)	
		CDKN1B (Del)	
Thyroid		YWHAE (T)	ABR-YWHAE
		PTPRD (Del)	CDK12-CSF3
			RAD18-SRGAP3
			SHROOM3-AFF1
Liver	VTI1A (T)	RMI2 (T)	VTI1A-NHLRC2
	MAP3K13 (T)	NCOR (T)	C3orf70-MAP3K13
	MACC1 (T)	CBLC (T)	AC005062.1-MACC1
	NSD3 (T)	MSH2 (T)	NSD3-AC087623.2
			RASGEF1B-VTI1A
			RMI2-TOX3
			NCOR1-LRRC75A
			MSH2-CYP3A43
Lung	CTNND2 (Del,T)	PTPRD (Del)	CTNND2-TRIO
	IKBKB (T)	RAD51B (T)	DUSP10-CTNND2
		FUS (T)	IKBKB-FAM91A1
		LRP1B (T)	FUS-CNOT1
			PDE6D-RAD51B
			PRKCH-HIF1A
			GAS7-LYRM9
			EHBP1-LRP1B

T, translocation; Dup, duplication; I, insertion; Del, deletion.

## Data Availability

Primary Bionano Saphyr data are available on request (jbroach@pennstatehealth.psu.edu).

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
