# Peer review of "Identification of Somatic Structural Variants in Solid Tumors by Optical Genome Mapping"

_jpm, 2021, doi:10.3390/jpm11020142_

Round 1
Reviewer 1 Report
The manuscript by Goldrich et al., on “Identification of Somatic Structural Variants in Solid Tumors By Optical Genomic Mapping” is well written and demonstrates the feasibility and reproducibility of detecting structural variants in solid tumors, using a magnetic bead-based protocol for the isolation of ultra-high molecular weight DNA from solid tumors. The study is important as it shows that optical genome mapping can be performed for solid tumors to detect SVs, which was previously limited to liquid tumors only.
However, the authors could include the following minor revisions:
- The authors have discussed reproducibility with triplicate and duplicate analysis from separate isolates. Although the comparison is provided elaborately, the authors can consider running few samples from the same isolate (DNA) on a different instrument for reproducibility studies. Perhaps the samples 7528 and 7708 with coverage <260 X and samples 3717 and 14369 with maximum coverage.
- The authors have listed the “Identification of cancer gene mutations” section in results where they provide events affecting cancer-related genes. Although, the manuscript is majorly methodological if discussing this section-the author can compare it to the clinical genetic diagnostic reports for these patients and provide additional information that OGM identified missed previously. As the manuscript is purely methodological, if this information is not available it should not preclude the acceptance of the manuscript.
Author Response
We thank the reviewer for the kind comments. With regard to the suggested minor revisions:
1) We have performed multiple Saphyr runs on the same DNA preps of a number of different samples (technical replicates) and find >95% reproducibility. We will include that number in the final version.
2) This is a great idea but unfortunately genomic evaluations are not routinely done on the types of tumor samples from patients at Hershey that comprised half our samples, so that comparative information is not available; nor was it available for the commercial samples. In other studies with liquid tumors, particularly AML and ALL, we have shown that Bionano returns a very different set of mutations than does targeted genome sequencing or even whole genome sequencing, particularly with regard to the spectrum of deletions. Bionano finds larger deletions that are routinely missed by whole genome sequencing while whole genome sequencing returns smaller deletions below the level of detection by Bionano. There is very little overlap between the two methods for deletions. We've also examined the overlap between Bionano and other methods such as karyotyping and CGH. In our study of 100 AML samples, Bionano finds everything those two methods find plus additional SVs and at higher resolution and with more information content. We can add references to those studies, which are published or available on MedRxiv.
Reviewer 2 Report
In general, the manuscript is unbalanced and many modifications should be done. Just one out of many examples: even though you paid huge attention to writing details in case of DNA extraction protocol, you said just a few about it in results and discussion. All comments are added directly to the text in the attached file.

Author Response
The attached file contains our manuscript revised in response to the comments of reviewer 2 as well as a copy of the reviewer's annotated version of the initial manuscript on which I've responded to each comment as a reply to that comment. The revised manuscript is included since we revised it extensively, including new figures and some reorganization of the presentation of results, to show the changes in the manuscript we performed to comply with the reviewer's suggestions.
